# OpenReview forum: "Training Socially Aligned Language Models on Simulated Social Interactions"
_ICLR.cc/2024/Conference — ICLR 2024 poster_

### Official Review · Reviewer_BAdV · 2023-10-31

**Soundness:** 2 fair
**Presentation:** 4 excellent
**Contribution:** 3 good
**Rating:** 6
**Confidence:** 3

**Summary:**

The paper proposes a framework for aligning LMs with societal norms by training them on data generated by LMs interacting with each other and correcting their behavior. They conduct extensive experiments showing how their methods improves alignment on a number of benchmarks, including human evaluation.

**Strengths:**

1. The paper proposes a quite original approach to alignment, using social interactions between multiple LMs as a data-generating process for finetuning. It’s ambitious and inspiring.
2. I think multi-agent experiments with LMs studying norm establishing, norm following and social learning are interesting on their own and the paper (specifically, the Sandbox environment) is a great contribution to this area.
3. The paper is well-written. Despite complexity of the method, it’s relatively easy to understand.
4. The experiments are quite extensive and well-designed. The results look promising: significant improvements can be seen on multiple relevant benchmarks.

**Weaknesses:**

1. I don’t think the authors have enough evidence to claim that “their approach is considerably more scalable” (from the abstract). I’m a bit concerned that the method does not scale with model size (page 4) and I didn’t see experiments showing scaling wrt society size. If it doesn’t scale, Stable Alignment is significantly less promising as an alignment method. (It’s still interesting as an LLM capabilities analysis though.)
2. I’m not entirely convinced how important is the social aspect of the data-generating process. It seems to be just an extension of [language model cascades](https://arxiv.org/abs/2207.10342) line of work, including [imitational learning from language feedback](https://arxiv.org/abs/2303.16755) (ILF) or [constitutional AI](https://www.anthropic.com/index/constitutional-ai-harmlessness-from-ai-feedback) (CAI), where LMs give feedback on LMs’ initial responses which are then refined. The agents are weakly individuated in the proposes framework, i.e. (as far as I understand) they start the same and and only differ in their conversation histories. I wonder how important is treating multiple calls to the same LM as simulating different agents. Does it improve alignment compared with a single-agent society, more similar to ILF or CAI? Is there a scaling plot? (I might’ve missed this, I’m happy to corrected by the authors.)
3. Relatedly, I think [language model cascades](https://arxiv.org/abs/2207.10342) line of work (including [imitational learning from language feedback](https://arxiv.org/abs/2303.16755) (ILF), [constitutional AI](https://www.anthropic.com/index/constitutional-ai-harmlessness-from-ai-feedback) (CAI), [critiques](https://arxiv.org/abs/2206.05802), [Reflexion](https://arxiv.org/abs/2303.11366), [STaR](https://arxiv.org/abs/2203.14465)) could be discussed briefly in Related work. I’m also happy to be corrected if the authors think I’m wrong seeing their paper as a multi-agent extension of this line of work.

**Questions:**

1. How is perplexity in Figure 5 measured, i.e. w.r.t. what data?
2. Fig 1c contains a typo: “Simiulated”

---

> ### Author Response · Authors · 2023-11-23
> **Thanks for your review and great suggestions!**
>
> Thank you for your detailed review and constructive feedback! We appreciate your recognition of our approach's novelty and the significance of our contribution. Below, we address your concerns and questions:
>
> **Weakness 1: Scalability of Stable Alignment**
>
> As explained on page 6, the scalability of Stable Alignment stems from its ability to decouple data collection from the training process. Unlike RLHF, which requires sequential pipeline operation with at least two large models in memory (see Figure 2 of the Instruct-GPT paper [1], which clearly shows sequential stages), Stable Alignment allows offline, asynchronous data production.
>
> This enables deploying hundreds or thousands of agents in the SandBox to interact in parallel. Our scaling analysis (Figure 3) demonstrates that societies with more capable LMs require fewer interactions to reach Pareto Optimality. Furthermore, we conducted additional experiments with larger LMs like Google PaLM 2 M/L and LLaMA 2, and the results demonstrate Stable Alignment can offer consistent improvement for LMs in different sizes (see results in Table A1).
>
> We hope all above can serve as evidence that Stable Alignment can perform well in scaling-up scenarios (data, model, number of interactions, etc.). Thanks again for your question!
>
> **Weakness 2: Importance of Multi-Agent Setup in Stable Alignment**
>
> We explored the necessity of the multi-agent setup by comparing it with a "single-teacher" approach ((i.e., not using the multi-agent setup, but distilling data from only one teacher model). The results show that a single-teacher setup leads to inferior performance (Table A2), lacking the self-improvement achievable through extensive agent interactions in a multi-agent environment. Training with data from a single teacher also resulted in slower and less effective alignment learning compared to our multi-agent approach (Figure 5b).
>
> **Weakness 3: Inclusion of More Related Works**
>
> We acknowledge your recommendation to discuss related works in language model cascades, such as imitational learning from language feedback (ILF), constitutional AI (CAI), critiques, Reflexion, and STaR. We have cited these works in the revised version of our paper and briefly discussed their relevance within the constraints of our page limit.
>
> **Questions: Perplexity Measurement and Typo Correction**
>
> For perplexity calculation, we used the LLaMA 70B model on generated samples (with the [script](https://huggingface.co/docs/transformers/perplexity)). Also, we have corrected the typo in Figure 1c in our revised manuscript.
>
> Finally, we feel grateful for your insightful review and are committed to continuously improving our paper. Please let us know if there are any further aspects we can enhance.
>
>
>
>
>
>
> [1] Training language models to follow instructions with human feedback

---

### Official Review · Reviewer_3QcT · 2023-10-31

**Soundness:** 4 excellent
**Presentation:** 3 good
**Contribution:** 3 good
**Rating:** 8
**Confidence:** 3

**Summary:**

This paper studies how to use simulated social interactions to align AI with human values. It presents a sandbox where many LLMs can interact with each other, provide feedbacks, and rate each other. Then a contrastive preference optimization, i.e. learn more from the welcome responses and unlearn the unwelcome responses. Experiments show that the proposed method gives better aligned responses compared to baselines.

**Strengths:**

1. The proposed method is interesting and conceptually novel, especially the sandbox idea.

2. The experiment is extensive (although contains some issues that I am concerned with, which I will discuss later)

3. The writing is clear.

**Weaknesses:**

The main issue that I am concerned with is the relation between this work and RLAIF[1]. It is also trying to use AI feedbacks to align the model. The main difference between the proposed method and RLAIF is that RLAIF still follows the paradigm of RLHF and only replace human feedback with AI feedback. And the proposed method is building up a sandbox of multiple AI. I am wondering how much difference in performance will be brought by this difference.


[1]Lee, Harrison, et al. "Rlaif: Scaling reinforcement learning from human feedback with ai feedback." arXiv preprint arXiv:2309.00267 (2023).

**Questions:**

See the weakness.

---

> ### Author Response · Authors · 2023-11-23
> **Thanks for your review and great suggestions!**
>
> Thank you for your comprehensive review and acknowledgment of the novel aspects of our work. Your insights are invaluable, and we've addressed your concern regarding the comparison with the RLAIF method as follows:
>
> **Weakness: Comparing Our Method with RLAIF**
>
> We appreciate your interest in comparing our Stable Alignment method with the RLAIF approach. Although both approaches share a similar foundation in using AI feedback for alignment, there are distinct differences in their applications and scope.
>
> RLAIF, released on September 1st, 2023, primarily targets summarization tasks and evaluations. In contrast, our Stable Alignment method, released on May 23rd, 2023, is more broadly focused on aligning AI with human values across a wider range of tasks and benchmarks, such as HH, Moral Stories, MIC, ETHICS, and TruthfulQA.
>
> Due to the unavailability of RLAIF’s code, a direct comparison is not feasible. However, we believe that some of our baseline methods and ablation settings offer a reasonable approximation of RLAIF's performance. For instance, TRLX, an open-sourced RLHF method, serves as a comparable model. Our findings indicate that TRLX, while valuable, does not match the stability or effectiveness of Stable Alignment (as detailed in Table 2 and Figure 5b). Similarly, our “learning from single teacher” baseline (i.e., not using the multi-agent setup, but distilling data from only one teacher model), also falls short of Stable Alignment's performance (see Table A2 and Figure 5b). We hope these comparisons provide a useful benchmark for estimating RLAIF’s performance in the absence of its official implementation.
>
> In our revised manuscript, we have cited the RLAIF work to acknowledge its relevance and contribution to the field.
>
> Thank you once again for your constructive feedback and encouragement! We hope our response clarifies the distinction and potential advantages of our Stable Alignment method in comparison to RLAIF.

---

### Official Review · Reviewer_ysok · 2023-11-04

**Soundness:** 3 good
**Presentation:** 3 good
**Contribution:** 3 good
**Rating:** 6
**Confidence:** 3

**Summary:**

The paper proposes a scalable and efficient training paradigm that enables LMs to learn from simulated social interactions. There are two key components: (1) simulating social interactions in SANDBOX with Back-Scatter to generate alignment data and (2) a 3-stage alignment learning framework (Stable Alignment) to train LMs on the alignment data from (1). Evaluation results on several relevant alignment benchmarks show the efficacy of the method.

**Strengths:**

-The proposed alignment method is novel, and takes inspiration from recently published research. The human evaluation results confirm experimental results on relevant benchmarks. Relevant related work is cited. The paper is well written, and graphics help with understanding.

**Weaknesses:**

-The proposed methodology is complicated compared to existing alternatives, but ablation shows that each component is critical in the effectiveness. I would assume the proposed method generalizes to other (larger) LMs, but additional experimental results would be helpful.

-I might overlook this but an analysis of the results on chatgpt will be helpful to identify future improvements.

-Experiments can be extended to more human value relevant datasets such as mmlu and civil comments.

-See questions below.

**Questions:**

A few clarifications on Back-Scatter would be helpful.

-It’s not clear how the observer LLMs generate the ratings using the Likert scale; the relevant prompts would be helpful given these ratings play a key part in the training component.

-Is the rating on imitation and self-critic data only alignment ratings.

-In Fig 3, are these the central agents? What is serving as the observer agent here?

The training process (3.2 Stable Alignment) is quite clear, though a few small clarifications would be helpful in the experiments section.

-Is instruction-tuning a prerequisite for alignment training with Stable Alignment?

-Could a non-instruction tuned model be trained on the alignment data and achieve comparable results (i.e. base LLaMA instead of Alpaca)?

-Do these experimental results on alignment tasks generalize to other (larger) base models?

-The wider performance implications are also unclear. Does model perplexity greatly increase? Does the performance on more generic tasks (non-alignment based benchmarks) decrease after alignment training with Stable Alignment?

Minor comments (not-relevant to rating):

-Naming: ChatGPT->GPT 3.5

-Related Work (Social Simulation) sentence 5: SandBox -> SANDBOX

-Fig 2 (upper left): Back Scatter -> Back-Scatter

---

> ### Author Response · Authors · 2023-11-23
> **Thanks for your review and great suggestions!**
>
> Thank you for your insightful review and constructive comments on our paper. We're glad you find our method novel and the evaluation convincing. Here's how we've addressed your points:
>
> **Weakness 1: Experimentation on Larger Language Models**
>
> We appreciate your suggestion regarding the need for experiments on larger LMs. In response, we conducted additional experiments with LLaMA-27B, Google’s PaLM 2 M (Bison), and PaLM 2 L (Unicorn) to examine the scalability of Stable Alignment. The results, which we'll add to the main paper, are in the new Appendix Section A.6 and Table A1. They indicate that Stable Alignment is more beneficial for smaller models, with a greater gain observed in the 7B models compared to the PaLM 2 M/L models. Furthermore, we noted improved performance with more powerful base models (LLaMA 2 outperforming LLaMA 1).
>
> **Weakness 2: Comparative Analysis of ChatGPT and Stable Alignment**
>
> Thank you for suggesting a detailed analysis of ChatGPT versus our Stable Alignment-trained LLaMA model. We revised the analysis section, adding a paragraph focusing on their differences. Despite their size disparity, this comparison has yielded actionable insights for future improvements in LM training methodologies.
>
> **Weakness 3: More general-purpose evaluation, such as MMLU?**
>
> We took your suggestion to extend our evaluation to more datasets seriously. In the new Appendix Table A2, we included results from MMLU for our default setting and some ablation settings with the recent LLaMA 2 model. The results show moderate improvement in zero-shot MMLU performance, likely due to the inclusion of societal topic discussions during training, which enriched the knowledge base.
>
> **Question 1: Prompts/Ratings details in SandBox Simulation**
>
> Yes, we have included the exact prompts we used throughout the simulation in the revised Appendix Section A.4.
>
> About ratings, we actually generate both alignment ratings and engagement ratings and filter the training data for imitation by using both ratings (e.g., we only choose to imitate responses whose alignment > 5 and engagement > 5). For self-critic, we teach the model to rate responses only in terms of alignment (so the model needs to generate the alignment rating + rationale), because the revisions used in the third step Realignment are only based on improvement in alignment.
>
> The points of ratings in Figure 3 are actually the average of each round (each round means you finish traversing all 100 agents). The observer agents in these scenarios are memory-less LMs like GPT-4. If you are interested in more details, please refer to our submitted code and data files. Thanks for the great question!
>
> **Question 2: Is instruction tuning a prerequisite for Stable Alignment?**
>
> Addressing your question about instruction-tuning as a prerequisite, we performed additional ablations on models with and without instruction tuning. Results in the newly added Table A1 indicate that training on base LLaMA models yields comparable results to those trained on instruction-tuned models. This suggests that the interaction data itself is a rich source of instruction-focused training, especially in alignment contexts. Our experiments with various other LMs also support the generalizability of Stable Alignment.
>
> And we believe our new experiments on other LMs (e.g., LLaMA-2, and PaLM 2 M/L) can already answer your questions about LM generalizations of Stable Alignment.
>
> Finally, we have corrected the typos you highlighted. We hope these revisions satisfactorily address your questions and concerns, and please let us know if anything else we can revise to make the work stronger!

---

### Official Review · Reviewer_MNpe · 2023-11-05

**Soundness:** 3 good
**Presentation:** 3 good
**Contribution:** 3 good
**Rating:** 6
**Confidence:** 3

**Summary:**

This paper proposes a data collection environment (Sandbox) and a training approach (Stable Alignment) to help align LMs to generate responses that are more moral and robust to adversarial attacks (i.e., being more "socially aligned").

The authors first introduces a new sandbox environment for collecting responses that are more socially aligned, by interacting a central agent (to be trained) with other static agents (e.g. LLMs). The other agents provide feedback and ratings to the answers generated by the central agent, and those data are later used for training. Next, the author proposes Stable Alignment, which trains a LM to learn from those collected data in three stages by 1) imitation learning from socially aligned responses, 2) learning to self-critique, and 3) learning to generate better responses entailed by a feedback. Additionally, stage 1 and stage 3 are trained using a contrastive objective. The authors finally showed improved performance (e.g. generating more human-aligned responses, choosing more moral responses) in a variety of tasks, and presents analysis showing that Stable Alignment has a more stable training curve compared to RRHF and requires less training steps to obtain high rewards/scores.

**Strengths:**

1. The proposed Contrastive Preference Optimization is new, and is shown to be able to effectively train LMs to learn from both positive samples (well-aligned responses) and negative ones. This hints at how data collection procedure in the future may not always need to focus on collecting high-quality, socially aligned data only, which can be costly and time-consuming.

2. The authors provided extensive analysis across six benchmark tasks with seven related baselines to show that Stable Alignment can generate/choose more socially aligned responses. This is helpful especially because the pipeline is autonomous: it does not require human involvement for additional annotation.

**Weaknesses:**

1. The authors claim the proposed Stable Alignment addresses limitations from (e.g.) RL from a reward model, which "may be inherently imperfect and not fully capture nuances of human judgment". However, since the data collection process (i.e., the sandbox) mainly prompts LLMs to gather feedback and ratings, how are these not imperfect or can be guaranteed to fully capture nuances of human judgement? There is a lack of analysis or discussion on the noises that could come from gathering data from LLMs, which is arguably critical for the proposed method to succeed and can be seen as limitation of how far this approach can go.

2. the sandbox data creation procedure appears to be an extension to prompt *multiple* LLM agents to provide rating and feedback for a single agent, instead of just prompting a single LLM teacher to gather data. There is a lack of experiments showing the advantage of using "multiple" LLM agents, and whether if this is even necessary. If not, many relevant work ([1]-[3]), which also did learning to generate feedback/critiques/improvement from a single teacher model, and is not discussed in this work. This limits the novelty and usefulness of the proposed "sandbox" construction and the following supervised training procedure.

**Questions:**

1. RL + reward model aims to train an LM with on-policy data. By converting the learning task of an offline scenario, the model eventually is training on *off-policy* alignment data, as its safety capability of the LM improves but it is still learning from the same fixed pool of data. The benefit of this is a more stable learning curve (from supervised training), but to what extent do you see this trade-off between on/off-policy training coming up in your proposed approach?

2. why is TRLX not included in Figure 4?

3. the motivation of sandbox being a "10 x 10" grid is unclear. Since the interaction mechanism is getting feedback and ratings from some other LLM agents, just sampling those LLMs from some unordered pool should suffice? It is unclear how the additional structure of a grid world is necessary in the proposed sandbox environment.

4. the data collection process of using LLM to gather feedback and asking the central model to generate a revision may be error-prone as indicated by [2] and [4], as a smaller LMs such as LLaMA-7b or Alpaca was not trained initially to follow the feedback and revise its response. Has this been an issue in the proposed approach as well? If so how is this resolved?
\
\
Some relevant work:

[1] Saunders, William, et al. "Self-critiquing models for assisting human evaluators." arXiv preprint arXiv:2206.05802 (2022).

[2] Seonghyeon Ye, et al. "SelFee: Iterative Self-Revising LLM Empowered by Self-Feedback Generation" https://kaistai.github.io/SelFee/ (2023).

[3] Welleck, Sean, et al. "Generating sequences by learning to self-correct." arXiv preprint arXiv:2211.00053 (2022).

[4] Yu, Xiao, et al. "Teaching Language Models to Self-Improve through Interactive Demonstrations." arXiv preprint arXiv:2310.13522 (2023).

---

> ### Author Response · Authors · 2023-11-23
> **Thanks for your review and great suggestions! (1/2)**
>
> Thank you for your thorough review and constructive feedback! We are glad you find our method novel and effective, and our evaluation extensive. Below we address your questions and concerns.
>
> **Weakness 1: Issues of reward-based optimization for alignment?**
>
> Thanks for the question. This sentence is a summation of the findings from recent studies (e.g., Chain-of-Hindsight, DPO, etc. all replace reward modeling with “verbal reward” or “directly learning from text”), which claim the reward-from might not be fine-grained enough to capture diverse and complicated human preferences during social interactions (i.e., our words “fully capture nuances of human judgement”). Also, there are some discussions on the intrinsic limitations of using a proxy model to delegate human judgment, such as reward gaming/hacking, and vulnerability under adversarial attacks (see the citations in the Introduction Section). These observations motivate us to propose doing rehearsal (i.e., thorough simulation of social interactions) in a separate space (i.e., the SandBox) to expose the issues, and let the LM learn from the corrections (i.e., the three stages of training in Stable Alignment).
>
> In the newly added Appendix Section A.6 and A.7, we conducted additional scaling analysis and ablation studies to understand how much we can obtain and how far we can go with Stable Alignment (results are mainly shown in the Table A1 and A2). In general, we find that Stable Alignment can bring more gains to smaller base LMs, and the gain is not sensitive to the type of the agent LM (RLHF’ed or not). In the revised Ethics Section, we properly warn the readers of some limitations and applicable conditions of our method. We have also included data, code, and exact prompts we used during data collection and training, to help follow-up works reproduce our results.
>
>
> **Weakness 2: Multi-agent vs Single Teacher?**
>
> Thanks for your great question! We actually explored the single teacher setting, where we remove all the internal memory systems on the agents, so that their draft answer, revision, judgement, etc. are all stateless. In other words, agents in the so-called single-teacher mode have no mechanism to keep track of their self-improvement during interaction. For a fair comparison, we generated the same number of data samples as the default mode and trained the LM on that with Stable Alignment (the three stages are the same; the only difference is how the data is generated).
>
> As shown in the newly added Table A2, single teacher data leads to regressive performance, compared with the data generated from the default multi-agent setting. This is because, without self-improvement, the quality of the simulation data is actually capped by the initial alignment status of the agent LM, which is apparently suboptimal since we find all the agent LMs can self-improve in the SandBox simulation (see Figure 3). So the single-teacher setting is equivalent to training with data that lacks self-improvement trajectories. In terms of Figure 5 (b), we also find training on single-teacher data is less efficient than training on multi-agent interaction data.
>
> **Question 1: Trade-off of Online/Offline RL?**
>
> Thanks for the question! We agree with the reviewer that there is a trade-off existing in choosing online/offline RL, and we believe that the key advantages of offline RL are crucial and its limitation can be potentially mitigated by simple extensions.
>
> According to the summarization paper [1], the offline RL is more suitable in cases where 1) data collection is expensive (which we think is probably true for collecting human value judgment data), and 2) the domain is complex and effective generalization requires large datasets (which is also true as RLHF requires a lot of human feedback). To enable more stable and efficient alignment learning, we build an asynchronous data cache (i.e., the fixed pool of data, as you mentioned), and the data collection thread (i.e., the SandBox simulation) can run continuously without bothering the learning thread. Compared with online RL (e.g., reward model based RLHF), we argue that the pre-trained reward model is also a snapshot of previously collected data, so its supervision is also approximate of real human judgment. However, SA has the advantages of easy deployment and training stableness.
>
> For future studies, one simple extension of sandbox + SA could be that we run this pipeline for multiple rounds: The societal problems used for next-round sandbox simulation are those whose responses are still not aligned even with the trained model in the previous round. After several rounds, we expect the leftover questions to be fewer and fewer, since the pool of data and the generative LM are updated iteratively and alternately (i.e., the pool is no longer strictly "fixed").
>
> (to be continued in the next response)
>
> [1] Offline Reinforcement Learning: Tutorial, Review, and Perspectives on Open Problems, Levine et al.

---

> > ### Comment · Reviewer_MNpe · 2023-11-23
> >
> > Thank you for your detailed reply! I would like to clarity that "Weakness 1" intends to say that both training/using a reward model and prompting LLMs to gathering feedbacks are imperfect. Therefore, the concern is that claiming a prompting-related method (i.e., author's proposed approach) can address the limitation of being "inherently imperfect and not fully capture nuances of human judgment" (i.e. using reward+RL) may be too strong. Granted, using a verbal feedback can provide a denser signal than a single scalar reward (as you also mentioned), but it still doesn't address the problem of both methods being "inherently imperfect". In the same way that it is hard to obtain a perfect reward function, to my knowledge there is no perfect "feedback generation" model. However, I do think your added experiments in appendix is useful to know in this context. I hope you can mention part of this discussion in your final manuscript.

---

> > > ### Author Response · Authors · 2023-11-23
> > > **Thank you!**
> > >
> > > Thanks for your quick response! Yes there might be no perfect reward/feedback generation model, and what we are doing is essentially better approximate the true distribution of human value judgment (no matter via rewards or verbal feedback). We still believe there is no standalone reward model in the human mind to perform value judgment, and thus we think Stable Alignment which learns from past experience during rehearsal or simulation might be a more natural way to learn alignment (not to mention its easy-to-deploy advantage).
> > >
> > > We will definitely include these new results into the main body of our paper, once given more space. Appreciate your great suggestions!

---

> ### Author Response · Authors · 2023-11-23
> **Thanks for your review and great suggestions! (2/2)**
>
> **Question 2: TRLX not included in Figure 4?**
>
> Thanks for the question! We have now included TRLX results in Figure 4.
>
> **Question 3: Why grid world?**
>
> Thanks for the question! The motivation for the grid-world design is two-fold: 1) Grid world is actually a classic design in alignment research (see an example in [AI Safety Gridworlds](https://arxiv.org/pdf/1711.09883.pdf), published in 2017; and a blog ["Specifying AI safety problems in simple environments"](https://deepmind.google/discover/blog/specifying-ai-safety-problems-in-simple-environments/) from Google DeepMind), and its simplicity can avoid incorporating potential bias from geographic design; 2) It offers ease for us to calculate the distance between agents so that we control the number of activated agents around one central agent to be roughly a constant. In our code, we actually use a radius = 3 (≈ a 3x3 grid) to scan the nearby agents and activate some of them with a preset dropout rate. Please take a look at our code if you are interested in the actual implementation.
>
> In general, the design of the grid world is to obtain more generalization, which follows a protocol in the prior arts, and it also provides some practical benefits in calculation. Thanks again for your great question!
>
> **Question 4: Simulation with small models is error-prone?**
>
> Not sure whether I understand this correctly but [2] and [4] are both using LLaMA-7B as the base model agent/multi-agent, while our setting is using much larger models (GPT-2/3.5/4) as the professional experience provider. In our preliminary experiments we find initializing a grid world with 10x10 LLaMA-7B models cannot generate proper response, as these pre-trained models cannot follow instructions well and the relatively small size limits the diversity of their generation severely. That's why we switched to our current setting later (which is learning from the interaction between many more capable models).
>
> **More related work**
> Thanks for the suggested related works! We have cited them in our revised version!
>
> Finally, we appreciate the review's careful reading, great questions, and constructive suggestions! Please let us know if anything is missing and we would be happy to make additional revisions!

---

> ### Comment · Reviewer_MNpe · 2023-11-23
>
> Thank you for your detailed reply! I would like to clarify that both [2] and [4] identify the challenge that, unlike LLMs, smaller models such as LLaMA-7B can NOT perform self-revision to a meaningful extend WITHOUT some training (e.g., specifically on self-improvement related data). In your approach, it seems that the data collection process relies on the central agent already being able to revise its attempts based on feedbacks from other agents (e.g. larger LMs). While your grid world consists of large models providing feedbacks, the central agent you train is still the "released Stanford Alpaca checkpoint". Therefore "Question 4" is concerned with the limited capability of smaller models to be able to meaningfully revise its attempt (work [2] and [4]), which could be troublesome as this seems to be a critical component in your data collection process.

---

> > ### Comment · Reviewer_MNpe · 2023-11-23
> >
> > Despite a few misunderstandings, I believe the authors have addressed most of the important concerns with additional experiments in appendices, mores references, and other fixes in the main paper. I have increased the contribution score to 3. I hope the authors will include those changes as well as some discussions on these not fully resolved questions in their final manuscript.

---

> ### Author Response · Authors · 2023-11-23
> **Thank you! Some clarification.**
>
> Thanks for the quick response and raising the score! We will definitely include the revisions in the final manuscript.
>
> We want to clarify that the all the social agents used in this work are "larger LMs" (GPT-4, GPT-3, etc.), as we mentioned in the beginning of Section 4. The reason was pretty similar to the conclusion found in [2] and [4]: In our preliminary experiments, smaller LMs such as LLaMA 7B were NOT capable of generating consistent and coherent responses to the given requests. Even if they were instruction-tuned, the diversity of their generation was questionable, due to their limited capacity. That's actually the reason why we decided to switch to "learning from pro" plan later, which means our goal is still teaching LLaMA 7B about alignment, but the experience is actually from pro players --- the self-improvement trajectories of those larger and more capable models during the simulation/rehearsal.
>
> More details of our SandBox simulation are covered in the Appendix Section A.1, where we have defined many key components such as Social Agents (i.e., LLMs + internal & external memory system). So in general, our experimental settings actually do not contradict the conclusions in [2] and [4], but align with them. We hope some misunderstanding might be clear after this explanation.
>
> We will make these key details more salient in the final version. We thank the reviewer for the constructive suggestions and the introduction to these great related works!

---

### Official Review · Reviewer_hsFk · 2023-11-08

**Soundness:** 2 fair
**Presentation:** 3 good
**Contribution:** 3 good
**Rating:** 5
**Confidence:** 3

**Summary:**

This paper has 3 key contributions: an environment for simulating social consensus and feedback on LLM responses; a training method for making use of the data gathered from the framework; and a new loss formulation that adds a contrastive aspect to perefence optimization. The authors conduct extensive experiments across multiple datasets and models.

**Strengths:**

1. The language in the paper is mostly clear and easy to read.


1. The general concept of being inspired by social interactions is interesting, and there is the possibility of being able to train an aligned model without the need to host an online reward model. However, there are many details in the execution that cast doubt on the learning method’s efficacy. Additionally, the claim of simulating human society with no supporting references from the social science literature is too grandiose.


2. There are extensive experiments on multiple datasets.


3. Human annotators were used for evaluating the final output rather than simply using a language model as a shortcut.

**Weaknesses:**

1. > However, this method often yields models susceptible to adversarial attacks, like “jailbreaking prompting” (Subhash, 2023; Xu et al., 2021), due to limited exposure to misaligned data during training (Amodei et al., 2016). To address this, a more advanced technique, “reward modeling” has been proposed (Leike et al., 2018; Christiano et al., 2017).

Subhash, 2023 uses ChatGPT, a model that has likely been trained with a reward model, as an example of a vulnerable model and appears to contradict the second sentence stating that reward modeling addresses the vulnerabilities of SFT-trained models.

2. The claim of simulating human society with no supporting references from the social science literature is way too grandiose. The authors should improve the accuracy of their claim.

3. It is somewhat unclear how the OpenAI models fit into the picture; are 99 of the 100 models all simply one of the 3 OpenAI models and one of them the Stable Alignment model? If so, this seems like a form of distillation of the preferences/values from the OpenAI model, which was purportedly trained with SFT and RLHF. Given that this method claims to do away with the need for RLHF/RMs, experiments showing its efficacy should not rely on models trained with such methods. In other words, the current experiments do not convince me that Stable Alignment can replace RLHF on its own. For example, I would be much more convinced if all the agents were initialized from non-RM based models. Relatedly, while convenient, using the OpenAI API models for critical parts of the experiments makes them essentially non-replicable and rather non-reproducible, given the constantly changing model in the backend.

4. An additional nitpick is that the method claims to deviate from SFT-like approaches but uses an SFT-trained model as the starting point. I would like to see this addressed as well; what is the performance of Stable Alignment without using an SFT-trained model as the initialization?

5. Regarding CPO, since it is a new technique that claims to improve the efficacy of the overall method, there should be an ablation where it is replaced with the regular SFT loss or other alignment algorithms, such as perhaps DPO.

6. More details on the procedure and exactly which models and types of social agents were used to generate data for each learning stage are needed; this is still unclear to me after reading through the details and appendices multiple times. Are the numbers of examples for each stage in Fig 2 the total over all iterations, or per iteration? How many iterations were needed to arrive at the final model? Was the model trained on data from all three societies at each stage or were 3 separate models trained? If the latter, which model was used in the final evaluation?

7. Use of HH-A as an estimate of adversarial prompt robustness:
I went through the examples in the HHH dataset (there are typos where the last H is missing throughout the paper) and disagree with the use of the appended “misaligned” response as a dataset good enough for adversarial robustness evaluation. A better evaluation of adversarial prompt robustness would be to use a library such as [garak](https://github.com/leondz/garak) with known adversarial prompts used by the community.

**Questions:**

1. Given that the motivation was alignment to social values, how was diversity across demographics ensured in the user study? The user study is meant to demonstrate that models trained with this method are better aligned, but better aligned to whom? This is important to call out since regular practitioners will simply use the popular models of the day with the released toolkit without thinking critically about whose values they capture.

1. Additionally, given the use of simulation with LMs is touted as a benefit of this approach, what are the drawbacks of using this instead of collecting representative feedback (and training a reward model), whose values are the models implicitly aligning to with such simulation? These should be explicitly called out and elaborated in the limitations section but are currently briefly glossed over in the ethics section.

1. Given the unstable nature of RL training, how was hyperparameter optimization done for the baseline methods? Were you able to reproduce published results where their hyperparameters were optimized?

1. For the embedding model and social agents, is it necessary to use an API model? Why not use an open-source model, which makes the experiments more reproducible and replicable?

---

> ### Author Response · Authors · 2023-11-23
> **Thanks for your review and great suggestions! (1/2)**
>
> Thanks for reading our work carefully! We are glad you find our idea of simulating social interactions in a community environment interesting. Below we address your questions and concerns:
>
> **Weakness 1: More accurate citations and better logic**
>
> Thanks for carefully reading our paper and for the great suggestion! We have fixed the citation and added some new related works in the second paragraph of the Introduction, with some edits on the text to make sure the logic flow is consistent and accurate. Please take a look at the revised version. Thanks again for your suggestion!
>
> **Weakness 2: Social science support for simulating human society with LMs**
>
> Thanks for the suggestion! Actually, in our original Related Work section, we have already included several milestone works from social science, HCI, and AI, that can serve as support for LMs simulated human society (e.g., the PNAS work [1], using LM simulation to replicate social science studies [2], confirming the fidelity of LMs simulated human society [3], etc.). Following your suggestion, we now have revised the first paragraph of Related Work to better summarize these works so that the readers can better notice and understand the social science background of our work.
>
>
> **Weakness 3: “I will be more convinced if all the agents were initialized from non-RM based models”**
>
> Thanks for the question! Yes, as described in Section 4, the LMs used for simulation not only include RLHF’ed (or RM-based) models (e.g., text-davinci-003 and GPT-4), but also include non-RM-based models (e.g., text-davinci-002). More information about these models was detailed in [the statement from OpenAI](https://platform.openai.com/docs/model-index-for-researchers).
>
> In the newly added Appendix A.6 and Table A1, we have included results when SandBox is initialized with non-RM-based models (i.e., we only use the text-davinci-002 to run the simulation to obtain interaction data). In general, the improvement over the baseline is still significant, partially because the data from text-davinci-002 actually takes up a main portion of the final mixture of the interaction data (nearly 50%, and 72% in the realignment data). As text-davinci-002 is not a RM optimized model, we believe these results can serve as evidence that the effectiveness of Stable Alignment is relatively model agnostic (as the improved alignment presented in the interaction data can only be attributed to the multi-agent simulation).
>
> **Weakness 4: More ablations about SFT**
>
> Thanks for the question! We have compared the performance with/without an SFT stage before Stable Alignment, and the results are in the newly added Appendix Section A.7 and Table A.2. In general, we find there is no significant difference between SFT/no SFT Stable Alignment. The reason could be the learning objective of Stable Alignment already includes the SFT term, so the convergence on Stable Alignment can guarantee the trained model is at least as good as an SFT-only model.
>
> **Weakness 5: Comparison with DPO?**
>
> Thanks for the question! In the original manuscript, we have already compared Stable Alignment with DPO. Please take a look at Figure 4, Table 2, and the corresponding analysis in the paper.
>
> **Weakness 6: More details on SandBox and data**
>
> Thanks for the question! About the model for the social agents, as we described in Section 4, we used text-davinci-002, text-davinci-003, and GPT-4 to run the simulation. For the number of iterations, it took 5, 2, and 1 round(s) for the above three models to reach Pareto Optimality in terms of both alignment and engagement. The number of data shown in Figure 2 presents the total number of data samples we collected from the simulation with the three models. For the three training stages, the major difference is in how we construct the data from the interaction trajectories (as shown in Figure 2), and we use a mixture of the data from the three models to construct those samples (though for ablation purposes we can easily select any one of them alone as the only data source). The supplementary material contains the data, code, and handy scripts to reproduce our work.
>
> (to be continued in the next response)
>
> [1] Socially situated artificial intelligence enables learning from human interaction
>
> [2] Using large language models to simulate multiple humans and replicate human subject studies
>
> [3] Out of One, Many: Using Language Models to Simulate Human Samples

---

> ### Author Response · Authors · 2023-11-23
> **Thanks for your review and great suggestions! (2/2)**
>
> **Weakness 7: HH-A and other adversarial tasks**
>
> This is a great suggestion! We have run additional evaluations with the garak framework, and we picked the RealToxicPrompt as the task that seems to be most related to our goal. The results are shown in the newly added Appendix Section A.7 and Table A.2, which we find are mostly aligned with our other alignment evaluations.
>
> **Question 1: Diversity, and demographics of the human evaluators**
>
> We acknowledge the importance of demographic diversity in our user studies. The revised version now includes detailed demographic information (i.e., gender, race, ideology) of the participants in Section A.5 and Figure A.4. We also included additional discussions and warnings in the revised Ethics Section (sorry we don’t have enough space in the main body of the paper), to make potential readers aware of the limitation of our work (e.g., the limited coverage all possible value judgment systems, which seems to be inevitable for any current LM based alignment research).
>
>
> **Question 2: Why not collect representative feedback to train reward models?**
>
> Thanks for the question! We believe the setting described by the reviewer is actually the typical reward model based method, which seems to have reward gaming/hacking problems (as reported by many related works we cited in the Introduction and Related Work Section). Another challenge is reward model based methods typically require at least two models pinned in the memory to interact with each other, which might be a practical deployment challenge in low-resource scenarios. Additionally, sometimes directly collecting enough representative feedback from a social group is time-consuming and costly, while simulation with proper conditions seems to be able to serve as a quick and low-cost alternative in such cases.
>
>
> **Question 3: Hyperparameter for other methods**
>
> Thanks for the question! For the tasks where baseline methods happen to report the results in their papers, we directly took their reported results since we believe those should be the best they can achieve. For other situations, we just followed their official codebase to configure the hyperparameters of their method.
>
> **Question 4: Why not use an open-source model running simulation locally?**
>
> The main concern of using a locally serving model is because of the hardware bottleneck --- since we start 100 agents and all of them need to interact (draft + feedback + revise) with each other in several rounds of simulation, the estimated number of calls would prohibitively large for locally serving settings. We also use multithread parallelization to speed up the call --- a vanilla local setting will need a large number of model checkpoints pinned in memory to handle this high throughput. In other words, it's more like practical hardware limitation, and we'd love to switch to open-source models that can be run and served locally at scale (100+ checkpoints?) as it will greatly save money cost of our experiments.
>
> Finally, thanks for the careful reading, great questions, and constructive suggestions! Please let us know if anything is missing and we would be happy to make additional revisions!

---

> ### Public Comment · ~StableAlignmentAuthors1 · 2023-11-27
> **Thank you! Some clarifications on crucial points.**
>
> Thanks the reviewer for the follow-up comments (sorry we just found the updated review in the original post)! Since as authors we are not allowed to post comments or submit the revised paper any more, we post our clarifications on some crucial points here:
>
> **Comparing with DPO**
>
> If we understand the reviewer correctly, the comparison asked by the reviewer is to only replace the final CPO training objective with DPO, since the reviewer says "*I'm looking for an ablation where DPO/something similar _replaces _ CPO in the experiments, not a comparison of the entire system against just DPO*". --- that's actually what we did in the comparison. Specifically, we replaced CPO training with the ["step 3"](https://github.com/eric-mitchell/direct-preference-optimization#step-3-run-dpo) as described in the DPO codebase, and the training was based on the same interaction data used for Stable Alignment (so the data factor was controlled). As DPO has no multi-agent simulation part to produce fine-grained interaction data, we think the only comparable part should be the training objective. Hope it is clearer now!
>
> **Weakness 6 and Question 4: "Are you simply calling the same model a hundred times for each society?"**
>
> Thanks for the question! We believe there might be a crucial misunderstanding of our work. We'd love to clarify this here and also highlight it in our final version. As we have described in the paragraphs on page 4, each social agent has a dedicated memory system: the internal memory to keep track of history opinions, and the external memory to record feedbacks and ratings (more details can be found in Appendix Section A.1. Details of Sandbox). The agents equipped with the memories are thus clearly not "the same model", since their memory systems updated differently during the thorough social interactions. Their responses in the next round will be constrained by retrieved in-context past opinions of their own (details on this part can also be found in Appendix Section A.1). That's actually the reason why we can see a self-improvement trend shown in Figure 3 (without these memories the agents have no mechanism to learn from positive/negative peer feedbacks).
>
> In other words, our work is trying to show the power of learning from collective intelligence (which we believe is the core of social interactions), rather than simply "calling the same model a hundred times for each society". We will make this clearer in the final version. Thanks for your question!
>
> There are other comments which might require adding new paragraphs to the revised paper, which is not allowed in current stage unfortunately (although we are very happy to do so!). We have already taken the notes of your great suggestions and will further improve our paper in the final version.
>
> We thank the reviewer again for the thoughtful reviews and questions!

---

> > ### Comment · Reviewer_hsFk · 2023-12-01
> >
> > I would like to clarify that I have not, in fact, misunderstood the work as believed by the authors. In fact, the authors' response is exactly in line with my understanding: while each agent is unique in the sense that the input sequence to the model (comprising memory, ratings, prompt, etc) is different for each agent, the underlying language model weights (GPT-4, etc) are the same for all agents in the society, i.e., the weights of the model are not unique for each agent. My previous concerns relating to this point therefore remain.

---

### Author Response · Authors · 2023-11-23
**Summarization of our revisions**

We are deeply grateful to all four reviewers for dedicating their time and expertise to thoroughly examining our work. The feedback provided has been invaluable in identifying both the strengths and areas for improvement in our paper.

Across the reviews, there is consistent recognition of the paper's innovations and contributions. Below we briefly summarize our revisions for the rebuttal:

1. Making edits on the **Introduction Section**, to make the logic flow more convincing and better supported by the cited works.
2. Conducting a bunch of new experiments and including the results in the newly added **Appendix Section A.6 and A.7**, which are mainly about scaling (training on Google PaLM 2 M/L), and ablations (SFT, instruction-tuning, multi-agent setup, etc.).
3. Adding more details about human studies, such as demographics information (**revised Appendix A.5**), exact prompts (**revised Appendix A.4**), and analysis on ChatGPT generation (**revised Appendix A.8**).
4. **Adding quite a few citations** and corresponding discussions, as suggested by several reviews (thanks!).
5. Making edits in the **Ethics Section** to cover more potential limitations.

All the above revisions are colored in brown in the submitted PDF. Please take a look and let us know anything else we can work on to improve the paper further! Thanks again!

---

### Meta-Review · Area_Chair_dhaJ · 2023-12-05

**Metareview:**

This paper introduces a framework for aligning language models with social norms and human values through simulated social interactions. The paper contributes a data collection environment, Sandbox, as well as a three-stage training approach, Stable Alignment, which involves imitation learning from socially aligned responses, learning to self-critique, and learning to generate improved responses based on feedback. The authors also introduce a new loss formulation that adds a contrastive element to preference optimization. They conducted extensive experiments across multiple alignment benchmarks using different models and included human evaluation. During the reviewer-AC discussion, the reviewers recognized the significance of the paper’s contributions but also mostly agreed on the following concerns:

1. Misalignment between the motivation and implementation: The paper presents its framework as learning through social interactions, but in reality, it involves repeatedly calling the same model a hundred times (GPT-4, text-davinci-002, text-davinci-003), which resembles self-reflection more than interaction. The final model does not interact with other LMs but learns from generated interaction data, resembling learning as an observer rather than a participant.
2. The authors argue that their approach offers greater scalability but revealed constraints in scaling up locally, needing to perform inference on 100 LLMs in parallel. Hosting a hundred LLMs to collect training signals seems less scalable than hosting a reward model in addition to the model being trained.
3. Uncontrolled confounders appear present: The authors claim their method can replace RLHF, but the non-RM ablation is problematic as they use ChatGPT as an observer in the simulation, which is trained with RLHF and may leak "RM knowledge". The SFT ablation seems to have an uncontrolled confounder for the same reasons.
4. In response, the authors state that a 7B LLaMa model was "not capable of generating consistent and coherent responses to the given requests," attributing this to model size. This contradicts a line from the paper, stating, "transitioning from a 6.8B to a 175B GPT-3 model, despite a 20-fold increase in model size, does not yield significant improvement. This suggests ... even smaller models can deliver satisfactory alignment performance."

Considering these limitations, the reviewers found this submission quite borderline, but agreed that none of these limitations is fatal. These limitations were provided to the authors during the author-AC discussion, and the authors listed changes they plan to implement for the camera-ready paper to mitigate some of these issues. I am satisfied with their plans and therefore recommend accepting this paper.

**Justification For Why Not Higher Score:**

The paper is relatively borderline, due to the four reasons listed in the meta-review.

**Justification For Why Not Lower Score:**

Social alignment of language models is a highly relevant topic, and the paper makes significant contributions to it. While the paper does have some weaknesses (as identified in the meta-review, points 1-4), none of them are fatal.

---

### Decision · Program_Chairs · 2024-01-16

Accept (poster)